# Water Quality Retrieval from ZY1-02D Hyperspectral Imagery in Urban Water Bodies and Comparison with Sentinel-2

**Zhe Yang [1,2], Cailan Gong [1,*], Tiemei Ji [3], Yong Hu [1] and Lan Li [1]**

1 Key Laboratory of Infrared System Detection and Imaging Technologies, Shanghai Institute of Technical Physics, Chinese Academy of Sciences, Shanghai 200083, China
2 University of Chinese Academy of Sciences, Beijing 100049, China
3 Shanghai Hydrological Station, Xuhui District, Shanghai 200232, China
* Correspondence: gcl@mail.sitp.ac.cn

**Abstract:** Non-optically active water quality parameters in water bodies are important evaluation indicators in monitoring urban water quality. Over the past years, satellite remote sensing techniques have increasingly been used to assess different types of substances in urban water bodies. However, it is challenging to retrieve accurate data for some of the non-optically active water quality parameters from satellite images due to weak spectral characteristics. This study aims to examine the potential of ZY1-02D hyperspectral images in retrieving non-optical active water quality parameters, including dissolved oxygen (DO), permanganate index ($COD_{Mn}$), and total phosphorus (TP) in urban rivers and lakes. We first simulated the in situ measured reflectance to the satellite equivalent reflectance using the ZY1-02D and Sentinel-2 spectral response function. Further, we used four machine learning models to compare the retrieval performance of these two sensors with different bandwidths. The mean absolute percentage errors (MAPE) are 24.28%, 18.44%, and 37.04% for DO, $COD_{Mn}$, and TP, respectively, and the root mean square errors (RMSE) are 1.67, 0.96, and 0.07 mg/L, respectively. Finally, we validated the accuracy and consistency of aquatic products retrieved from ZY1-02D and Sentinel-2 images. The remote sensing reflectance ($R_{rs}$) products of ZY1-02D are slightly overestimated compared to Sentinel-2 $R_{rs}$. ZY1-02D has high accuracy and consistency in mapping $COD_{Mn}$ products in urban water. The results show the potential of ZY1-02D hyperspectral images in mapping non-optically active water quality parameters.

**Keywords:** urban water quality; non-optically active parameters; remote sensing; ZY1-02D; Sentinel-2

## 1. Introduction

Water resources have a significant role in different functions of cities, such as drinking water, industrial production, and landscape [1]. Rivers and lakes are an important part of urban water bodies. However, with the rapid development of urban society and economy, living needs grow rapidly, and the discharge of domestic water, agricultural water, and industrial water exceeded the self-cleaning capacity of water bodies, causing serious pollution to urban rivers and lakes [2]. Deterioration of urban water quality brings the safety issue of drinking water and destruction of the ecological environment, which in turn affects human health and biodiversity [3]. The city water department needs to regularly formulate policies based on water quality assessment data for the further development of the city.

Water quality monitoring is an important part of water quality evaluation. It aims to understand the water quality of urban water bodies, especially rivers and lakes. Although traditional water quality monitoring methods including manual field sampling and laboratory measurements or automatic in situ measurements have high accuracy, the manual method is labor-intensive, and the construction and maintenance of the automatic station requires expensive costs. Furthermore, both methods can only reflect the water quality

at specific sampling points; it is challenging to meet the requirements to monitor water quality over the entire water surface of rivers and lakes [4]. In comparison, remote sensing has been widely used to monitor water quality since the 1970s with temporal and spatial characteristics [5,6].

Dissolved oxygen (DO), permanganate index ($COD_{Mn}$), and total phosphorus (TP) are important chemical indicators for water quality monitoring. These parameters are called non-optically active parameters because they do not absorb light and are difficult to estimate directly from spectral characteristics [7]. DO is the amount of oxygen that is present in water and is impacted in complex ways by Chl-a and algae [8]. Although DO can absorb ultraviolet light, this only makes it measurable in the laboratory. $COD_{Mn}$ is the amount of potassium permanganate oxidant consumed in the treatment of water samples and reflects the concentrations of organic pollutants in water [9]. Phosphorus provides favorable conditions for algal growth [10]. For remote sensing, based on the strong correlation between optically active parameters and non-optically active parameters, indirect methods have been used to estimate or measure several of these important water quality parameters [11].

In previous studies, the researchers often carried out research using empirical methods to estimate non-optically active parameters [12–14]. Empirical methods typically train and calibrate a regression model between image-derived features and water quality parameter concentrations from in situ observations [15]. Therefore, empirical methods rely on using multispectral sensors because they have a good temporal and spatial resolution to obtain more ground matching to train models. For instance, Al-Shaibah et al. [16] built empirical algorithms between Landsat images and water quality (V-phenol, DO, $NH_4$-N, $NO_3$-N). Huang et al. [17] used the super-resolution algorithm and statistical regression models to retrieve $NH_3$-N, COD, and TP in small-sized rivers. Gao et al. [18] used band combinations and regional multivariate statistical modeling techniques to retrieve the TP from HJ-1A images in Chaohu Lake. For hyperspectral sensors, Chang et al. [19] and Merin et al. [20] used Moderate-resolution Imaging Spectroradiometer (MODIS) images to retrieve nutrient (TN or TP) concentrations, but the spatial resolution of most spaceborne hyperspectral sensors limits the application to small and medium-sized inland rivers and lakes. In addition, non-optically active parameters can be estimated by their correlation with other parameters such as chlorophyll-a, organic matter, etc. [21]. Lu [22] developed an indirect algorithm based on the correlation between TP and optically active parameters. However, the correlation between the non-optically active parameters and optically active parameters was not assured in different regions [23]. Instead of using image-derived features, a more general regression model can be trained and calibrated using a broad range of in situ observations, including optical properties and concentration of water quality [24]. Up to now, methods based on in situ observation modeling in the literature are mostly limited to the use of optically active parameters [25].

In this study, we retrieve the water quality parameters from ZY1-02D and Sentinel-2 images based on an in situ observation modeling method. We used four machine learning algorithms to train and calibrate models between simulated in situ reflectance and concentrations. The in situ observations-based inversion of water quality parameters requires precise atmospheric correction, which is not critical for image-derived-based methods. We assessed the consistency of the ZY1-02D-derived $R_{rs}$ and water quality products with those of Sentinel-2. The comparisons of the two sensors provide direct evidence for the potential of ZY1-02D hyperspectral imagery for retrieving water quality parameters.

## 2. Materials and Methods

### 2.1. Study Area

Shanghai is located in the delta region in the lower reaches of the Yangtze River, at $120°52'$–$122°12'$E longitude and $30°40'$–$31°53'$N latitude (Figure 1). Shanghai is densely covered with rivers and lakes, and it covers a water area of 649.2 km$^2$, with a river density of 4.79 km/km$^2$ and a surface water region ratio of 10.24% [26]. The main natural water

systems are the Yangtze River, Huangpu River, and Dianshan Lake, which provide most of the domestic water for inhabitants. As one of the largest cities in China with a population of over 24 million in the year 2020 [27], the rapid economic development and human activities have led to the deterioration of the water quality in Shanghai [28].

In this study, two typical and important rivers and lakes were selected for detailed evaluation and analysis. They are the Huangpu River and its upper tributaries and Dianshan Lake. Huangpu River is the largest river in Shanghai and flows through most of the districts. Dianshan Lake is the largest lake in Shanghai and is located in Qingpu District.

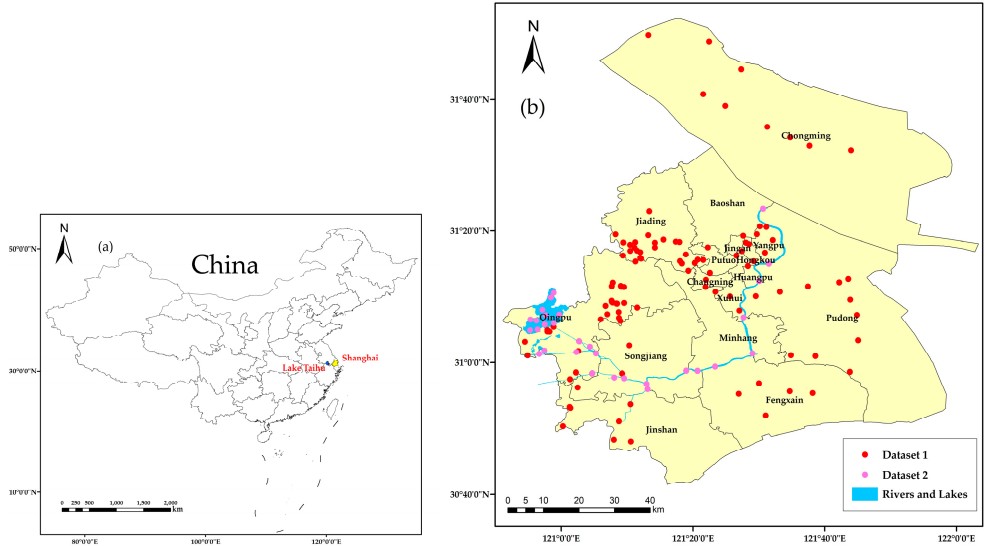

**Figure 1.** (**a**) Spatial distributions of Shanghai and Taihu Lake in China. (**b**) Study area and in situ sample distribution in Shanghai.

### 2.2. Materials

#### 2.2.1. In Situ Data

From September 2018 to November 2021, 12 remote sensing reflectance ($R_{rs}$) field measurements were carried out in 90 rivers of Shanghai. The average transparency of these rivers is lower than 0.6 m, so the reflectance from the bottom of the rivers cannot affect the $R_{rs}$ measurements of surface water [29]. The main goal of all measurements was to characterize the relationship between non-optically active parameters and optically active parameters by measuring as many rivers as possible at different dates and locations. The average width of these rivers ranges from 10 to 150 m. We tried to balance the number of rivers with good and bad water quality, and some rivers measured in 2018 were defined as black and smelly water bodies. Finally, a total of 183 sets of data were obtained. The measurements covered all seasons and nine districts in Shanghai (Table 1).

For each measurement, $R_{rs}$ was measured on the bank or bridge of rivers using a Fieldspec 4 spectroradiometer ranging from 350 to 2500 nm (1 nm interval). Referring to the above-water method and NASA-recommended measurement standards [30,31], specifically, measurements were performed between 9:00 and 15:00 on sunny windless days, and the zenith angle and the azimuth angle were 45° and 135°, respectively. Total water surface radiation ($L_{sw}(\lambda)$), skylight radiance ($L_{sky}(\lambda)$), and the reference plate radiance ($L_p(\lambda)$) were measured at each site. $R_{rs}$ was calculated by the following equation:

$$R_{rs}(\lambda) = \frac{L_{sw}(\lambda) - \rho_{sky}(\lambda)L_{sky}(\lambda)}{\pi L_p(\lambda)/\rho_p(\lambda)},$$ (1)

where the $\rho_{sky}$ is the air–water interface skylight reflectance and related to wind speed and solar altitude. According to field measurement conditions, we used 0.028 [31]. $\rho_p$ is the

irradiance reflectance of the gray plate (30%). The in situ measured $R_{rs}$ of these sampling sites are shown in Figure 2.

**Table 1.** Dates, locations, and sample number of each measurement.

| No. | Date | The District of Shanghai | Number |
|---|---|---|---|
| 1 | 19 September 2018, 26 September 2018, 27 September 2018 | Jiading | 37 |
| 2 | 17 December 2018 | Jiading | 12 |
| 3 | 12 April 2019, 17 April 2019 | Qingpu | 21 |
| 4 | 21 May 2019, 22 May 2019 | Jiading | 11 |
| 5 | 7 April 2021, 8 April 2021 | Qingpu, Pudong | 25 |
| 6 | 9 May 2021, 10 May 2021 | Changning, Jinshan | 15 |
| 7 | 1 June 2021, 6 June 2021 | Qingpu, Chongming | 12 |
| 8 | 6 July 2021 | Chongming | 5 |
| 9 | 5 August 2021, 9 August 2021 | Xuhui, Yangpu | 17 |
| 10 | 7 September 2021 | Pudong | 8 |
| 11 | 12 October 2021 | Hongkou | 12 |
| 12 | 13 November 2021 | Yangpu | 8 |
| Total | / | / | 183 |

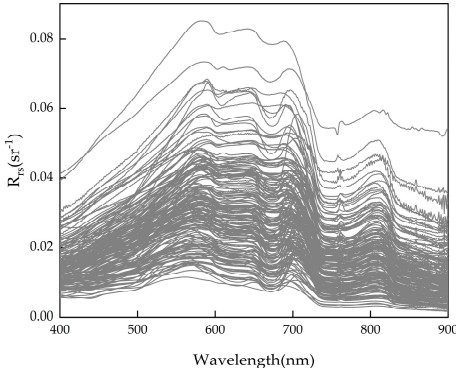

**Figure 2.** In situ measured $R_{rs}$.

The field-measured DO, COD$_{Mn}$, and TP concentration data were provided by Shanghai Hydrological Station. Table 2 shows the laboratory analysis methods and descriptive statistics of parameters. The dataset shows a large water quality parameter range. The DO, COD$_{Mn}$, and TP concentration measurements were in accordance with the GB 7489-87, GB 11892-89, and GB 11893-89, respectively.

**Table 2.** Laboratory analysis methods and statistics of DO, COD$_{Mn}$, and TP.

| Water Quality Parameters | Laboratory Measurement Methods | Mean | Min. | Max. | Std |
|---|---|---|---|---|---|
| DO | Iodometry method | 5.9 | 2.0 | 12.8 | 2.3 |
| COD$_{Mn}$ | Permanganate index method | 4.58 | 2.10 | 11.40 | 1.58 |
| TP | Molybdenum antimony spectrophotometry | 0.172 | 0.041 | 0.664 | 0.093 |

All data were divided into two parts (Figure 1b). Dataset 1 (183 samples) was concurrent with radiometric measurements used to develop the algorithm for water quality parameter estimation. Figure 3 shows the distributions of each parameter. All parameters were missing the high concentration part, and COD$_{Mn}$ and TP were more concentrated in low concentration areas. Dataset 2 (30 samples) was the near-coincident data obtained under Sentinel-2 and ZY1-02D overpasses used to validate the image-retrieved results.

Dataset 2 was mainly distributed in Dianshan Lake and rivers with a width greater than 100 m. According to our field investigation, the water change cycle of Dianshan Lake is 7 days and the flow rate of these rivers is slow. Therefore, in the absence of precipitation and sudden pollution, the water quality of these sites does not change much in a short period of time.

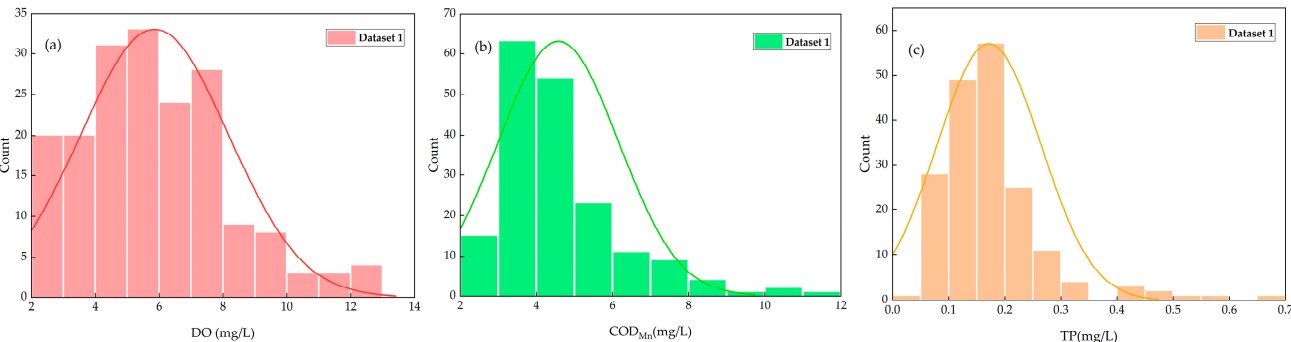

**Figure 3.** Distributions of Dataset 1's DO (**a**), COD$_{Mn}$ (**b**), and TP (**c**).

### 2.2.2. Independent Dataset in Taihu Lake

We also used in situ $R_{rs}$ and water quality concentrations (Dataset 3) from Taihu Lake in 2009 to validate the suitability of Dataset 1's models across locations and times. Samples were collected monthly at 32 stations in Taihu Lake (Figure 4). $R_{rs}$, DO, COD$_{Mn}$, and TP were determined using the methods described in Section 2.2.1. Overall, we selected 91 samples distributed across Taihu Lake between February and May.

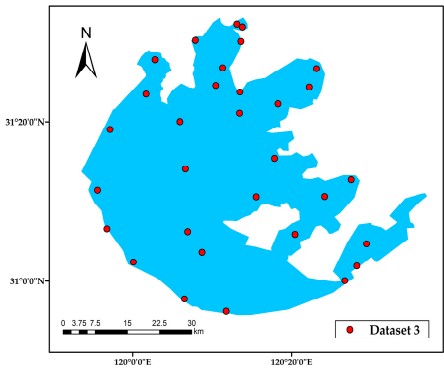

**Figure 4.** Locations of Dataset 3 collected in Taihu Lake.

### 2.2.3. Satellite Data

The multispectral data used in this study was a Sentinel-2 image with a spatial resolution of 10–60 m. The Multispectral Instrument (MSI) onboard Sentinel-2(A/B) has both 13 bands and short temporal resolution. The S2A-MSI Level 1C image of Shanghai acquired on 13 May 2020 was downloaded from the Copernicus Open Access Hub (https://scihub.copernicus.eu/ accessed on 5 December 2020).

Hyperspectral data with coincident Sentinel-2 images were acquired by the Advanced HyperSpectral Imager (AHSI) onboard the ZY1-02D satellite of China. The AHSI has 150 bands in the visible and near-infrared (VNIR) and 180 bands in the short-wave infrared (SWIR) with 10 and 20 nm spectral resolution, respectively [32]. Meanwhile, it has a spatial resolution of 30 m, which can meet the needs of urban water monitoring. In addition to Band 5 of MSI, AHSI can have at least 2 bands in each of the MSI band's configurations (Table 3).

Both the Sentinel-2 and ZY1-02D images have less than 10% cloud coverage and a time difference of ±3 days with Dataset 2.

**Table 3.** Band number, wavelength, and resolution of the MSI and AHSI used in this study.

| Sentinel-2A MSI | | | ZY1-02D AHSI | | |
|---|---|---|---|---|---|
| Band Number | Wavelength (nm) | Resolution (m) | Band Number | Wavelength (nm) | Resolution (m) |
| 1 | 433–453 | 60 | 6–7 | 433–452 | 30 |
| 2 | 458–523 | 10 | 9–15 | 459–521 | 30 |
| 3 | 543–578 | 10 | 19–22 | 545–581 | 30 |
| 4 | 650–680 | 10 | 31–34 | 648–675 | 30 |
| 5 | 698–713 | 20 | 37 | 700–710 | 30 |
| 6 | 733–748 | 20 | 41–42 | 734–753 | 30 |
| 7 | 773–793 | 20 | 45–47 | 769–796 | 30 |
| 8 | 785–900 | 10 | 47–59 | 786–899 | 30 |
| 8A | 855–875 | 20 | 55–56 | 854–873 | 30 |

### 2.3. Methods

#### 2.3.1. Satellite Band $R_{rs}$ Simulation

To develop retrieval models that can be used on MSI and AHSI images, the in situ $R_{rs}$ should be simulated to satellite band equivalent reflectance. To achieve this, convolution via Equation (2) based on the spectral response function (SRF) of MSI and AHSI sensors was calculated as follows:

$$R_{rs}(B_i) = \frac{\int_{\lambda_{min}}^{\lambda_{max}} R_{rs}(\lambda) SRF_i(\lambda) d\lambda}{\int_{\lambda_{min}}^{\lambda_{max}} SRF_i(\lambda) d\lambda}, \tag{2}$$

where $SRF_i(\lambda)$ is the relative SRF of the MSI and AHSI $i$th band, and the Gaussian function was used to describe SRF of the AHSI sensor [33]. $\lambda_{max}$ and $\lambda_{min}$ are the wavelength ranges in this band.

The MSI and AHSI equivalent reflectance spectra are displayed in Figure 5. Figure 5c shows the average of in situ $R_{rs}$ and the equivalent reflectance simulated by the two sensors. For the hyperspectral sensor AHSI, the average equivalent reflectance of each band and in situ $R_{rs}$ basically coincide. However, some bands' average equivalent reflectance of the multispectral sensor MSI deviates significantly from the in situ $R_{rs}$.

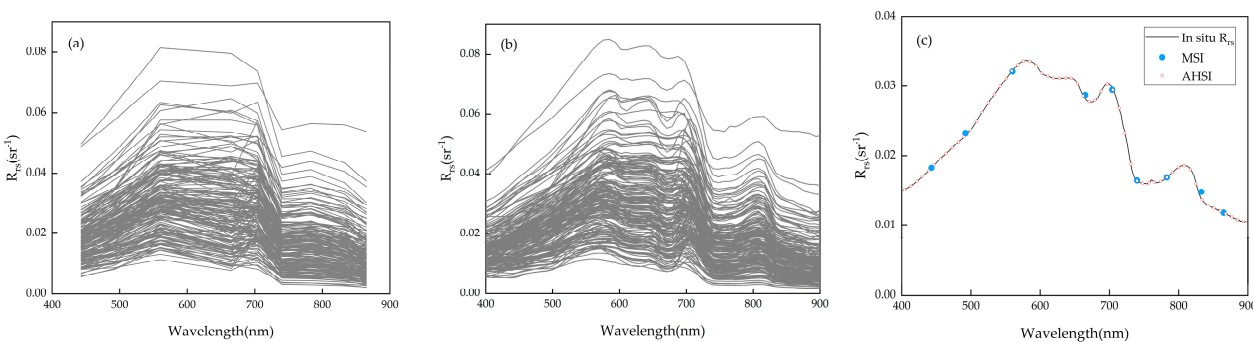

**Figure 5.** MSI (**a**) and AHSI (**b**) equivalent reflectance spectra and the comparison with average in situ reflectance (**c**).

#### 2.3.2. Model Development

The Pearson-based correlation analysis was used to describe the correlation between the water quality parameters and the satellite equivalent reflectances. The Pearson correlation coefficient ($r$) ranges from −1 to +1. When the $r$ is close to −1 or +1, this indicates a strong inverse or positive correlation between the variables, respectively. However, the $r$ close to zero indicates no correlation between the variables [34]. As shown in Figure 6, the

correlations varied significantly for different water quality parameters, but they are both at a low level (highest $|r|$ = 0.41).

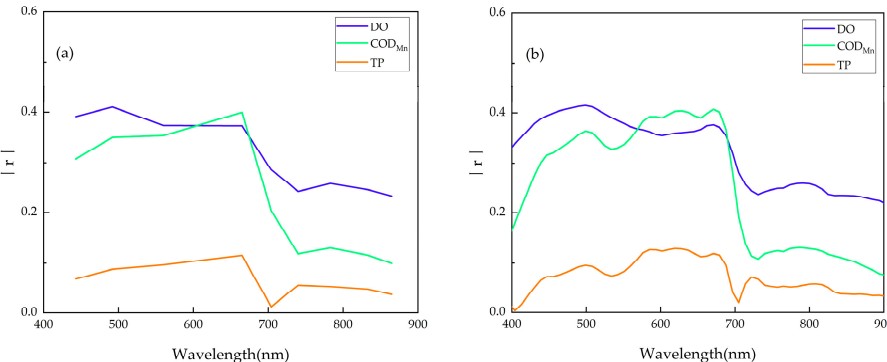

**Figure 6.** The absolute value of *r* between water quality parameters and MSI (**a**) and AHSI (**b**) equivalent reflectance.

In this study, there were four machine learning methods that were considered to develop retrieval models: support vector regression (SVR), partial least squares regression (PLSR), K-nearest neighbor (KNN), and XGBoost. SVR can solve the non-linear problems in low dimensional feature space by transforming the input data to a high dimensional space with a non-linear function, then seeking a linear regression hyperplane in high dimensional feature space [35]. PLSR model the covariance relations between features and targets by finding the latent variables, so it can reduce the multicollinearity among input values [36]. KNN predicts the target by local interpolation of the targets associated with the nearest neighbors in the training set [37], and XGBoost is a gradient boosting tree model, which predicts the sum of scores in multiple regression trees [38]. All methods were implemented by scikit-learn of Python.

### 2.3.3. Satellite Data Preprocessing

The main preprocessing of the satellite images included atmospheric correction (AC), water body extent extraction, and remote sensing reflectance calculation [39]. First, the radiometric calibration coefficients of AHSI were used to convert the digital number values to top-of-atmosphere radiances. Second, the Sen2Cor processor was used to obtain the Sentinel-2 Level 2A Bottom of Atmosphere reflectance product [40]. The FLAASH atmospheric correction module in ENVI was used to retrieve surface reflectance for AHSI images [41]. In this study, the mid-latitude summer atmosphere and rural aerosol were selected in FLAASH.

The modified normalized difference water index (MNDWI) and the OTSU [42] method were used to segment the water bodies from images. Based on the assumption that the minimum surface reflectance in the SWIR band of turbid water is composed only of residual aerosol scattering, skylight, and sun glint, a remote sensing reflectance estimation method [43] was used for MSI and AHSI surface reflectance images to correct the skylight effect and retrieve remote sensing reflectance as follows:

$$R_{rs}^c(\lambda) = \frac{R(\lambda) - \min(R_{SWIR})}{\pi}, \tag{3}$$

where $R_{rs}^c(\lambda)$ represents the remote sensing reflectance, $R(\lambda)$ represents the surface reflectance, and $\min(R_{SWIR})$ indicates the minimum surface reflectance of the SWIR band in MSI and AHSI, where $R_{SWIR}$ of AHSI use the average of R in the 1530–1630 nm bands.

2.3.4. Accuracy Assessment

The coefficient of determination ($R^2$), mean absolute percentage error (*MAPE*), and root mean square error (*RMSE*) were used to assess the performance of water quality retrieval models and the agreement between in situ data and image retrievals [44].

$$R^2 = 1 - \frac{\sum_{i=1}^{n}(M_i - E_i)^2}{\sum_{i=1}^{n}(M_i - \overline{M})^2},$$ (4)

$$MAPE = \frac{1}{n}\sum_{i=1}^{n}\frac{|M_i - E_i|}{M_i} \times 100\%, \text{ and}$$ (5)

$$RMSE = \sqrt{\frac{\sum_{i=1}^{n}(M_i - E_i)^2}{n}},$$ (6)

where $n$ is the number of samples, $M_i$ and $E_i$ represent the measured values and estimated values, respectively.

## 3. Results

### 3.1. Spectral Response to Non-Optically Water Quality Parameter Variation

The mean values of in situ spectra for DO, $COD_{Mn}$, and TP concentrations in different value ranges are shown in Figure 7. Overall, the spectral reflectance between 400 and 700 nm is inversely proportional to the DO concentration. In 2–8 mg/L and 8–12 mg/L regions, the spectral reflectance is similar from 700 to 900 nm. When the DO concentration is higher, the spectral reflectance trough at 675 nm and peak at 705 nm are more obvious. These indicate that for water with lower DO, suspended sediment accounts for the largest proportion, and for water with higher DO, the spectral variability as a response to Chl-a and algae is more obvious.

Similarly, the reflectance between 400 and 700 nm is inversely proportional to the $COD_{Mn}$ concentration. When the $COD_{Mn}$ concentration is greater than 3 mg/L, the spectral reflectance is similar from 700 to 900 nm. The water with $COD_{Mn}$ concentration over 6 mg/L is usually considered to be polluted. As shown in Figure 7b, the spectral reflectance peak at ~700 nm of high $COD_{Mn}$ concentration moves toward the longer wave.

In 0.1–0.4 mg/L regions of TP concentration, the spectral reflectance between 400 and 700 nm is inversely proportional to concentration, and the spectral reflectance between 700 and 900 nm is similar. When the TP concentration is lower than 0.1 mg/L, the spectral reflectance also shows the characteristics of a high Chl-a concentration, which indicates that in one water body the correlation with Chl-a might be with P or N. The spectral reflectance peak of high TP concentration is also obvious at 705 nm.

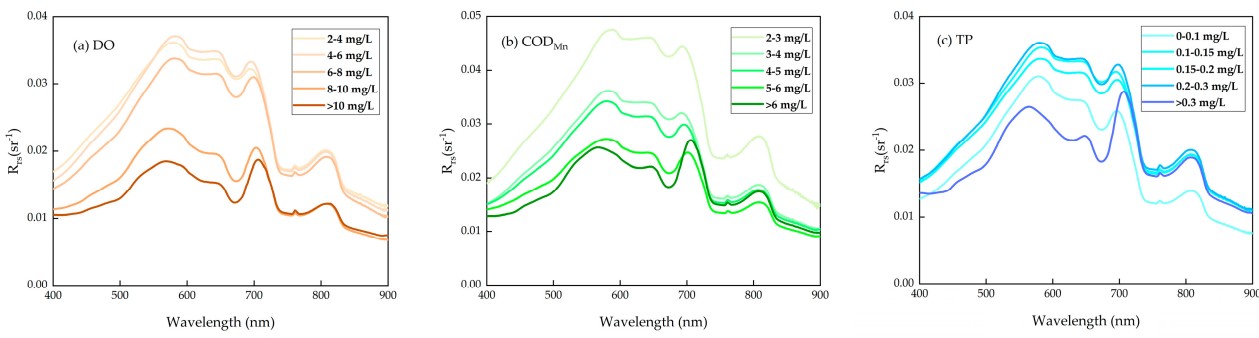

**Figure 7.** Average spectra of DO (**a**), $COD_{Mn}$ (**b**), and TP (**c**) in different concentration ranges.

### 3.2. Development and Validation of Machine Learning Models

3.2.1. Model Structure and Inputs

To compare the aquatic products retrieved from AHSI and MSI, the band selection range in this study was the nine visible and near-infrared bands of MSI and the AHSI bands of the corresponding wavelength range (Table 3). Although other bands of AHSI could be more suitable, we did not consider the bands out of range for the MSI sensor's spectral configuration. The input of each model included nine bands and three band ratios. Previous studies have confirmed that adding band ratios can improve the performance of water quality retrieval models [45–47]. The second correlation analysis between water quality parameters and equivalent reflectance ratios was carried out to find better model inputs. According to Figure 6 and Table 4, the most appropriate band composition of each water quality parameter was used to develop SVR, PLSR, KNN, and XGBoost models. For band ratios, all of the variables were significantly correlated at the 95% confidence level, with higher correlation coefficients compared to the single-band variables.

**Table 4.** The absolute value of *r* and *p*-value between water quality parameters and MSI and AHSI equivalent reflectance ratios.

| | Band Ratio | $|r|$ | | |
|---|---|---|---|---|
| | | **DO** | **COD$_{Mn}$** | **TP** |
| **MSI** | $R_{rs}(B_3)/R_{rs}(B_4)$ | **0.4 *** | 0.41 * | 0.12 |
| | $R_{rs}(B_5)/R_{rs}(B_4)$ | **0.38 *** | **0.66 *** | **0.42 *** |
| | $R_{rs}(B_6)/R_{rs}(B_4)$ | 0.25 * | **0.59 *** | **0.43 *** |
| | $R_{rs}(B_6)/R_{rs}(B_7)$ | **0.32 *** | 0.16 * | 0.02 |
| | $R_{rs}(B_7)/R_{rs}(B_4)$ | 0.21 * | **0.56 *** | **0.41 *** |
| **AHSI** | $R_{rs}(B_{22})/R_{rs}(B_{33})$ | **0.44 *** | 0.4 * | 0.06 |
| | $R_{rs}(B_{37})/R_{rs}(B_{31})$ | 0.35 * | 0.66 * | **0.44 *** |
| | $R_{rs}(B_{37})/R_{rs}(B_{33})$ | **0.4 *** | 0.66 * | 0.4 * |
| | $R_{rs}(B_{37})/R_{rs}(B_{34})$ | 0.4 * | **0.66 *** | 0.41 * |
| | $R_{rs}(B_{41})/R_{rs}(B_{33})$ | 0.29 * | **0.63 *** | **0.43 *** |
| | $R_{rs}(B_{42})/R_{rs}(B_{45})$ | **0.34 *** | 0.17 * | 0.03 |
| | $R_{rs}(B_{45})/R_{rs}(B_{33})$ | 0.25 * | **0.6 *** | 0.42 * |
| | $R_{rs}(B_{47})/R_{rs}(B_{34})$ | 0.24 * | 0.6 * | **0.42 *** |

* Significant at 5% probability, the band ratios selected for each parameter are marked.

To examine that adding variables to inputs is not just fitting noise, we tested the performance of XGBoost on the validation dataset using three combinations: (1) five bands without red-edge bands; (2) total nine bands; (3) nine bands and three band ratios. Table 5 shows that the input variables with 12 variables produced the best performance.

**Table 5.** Performance metrics of XGBoost model using three combinations.

| Parameter | Variables | $R^2$ | MAPE (%) | RMSE (mg/L) |
|---|---|---|---|---|
| DO | 5 | 0.21 | 29.98 | 2.18 |
| | 9 | 0.32 | 29.30 | 2.02 |
| | 12 | 0.53 | 24.28 | 1.67 |
| COD$_{Mn}$ | 5 | 0.38 | 21.56 | 1.33 |
| | 9 | 0.42 | 21.47 | 1.29 |
| | 12 | 0.65 | 17.99 | 1.0 |
| TP | 5 | 0.08 | 37.69 | 0.096 |
| | 9 | 0.27 | 39.88 | 0.086 |
| | 12 | 0.48 | 37.04 | 0.073 |

3.2.2. Performances of Machine Learning Models

Dataset 1 was randomly divided into a training dataset (N = 128) and a validation dataset (N = 55). For the development of the model, all machine learning models used

the same training dataset, and the hyperparameters were determined by the strategy of grid search. Comparing the water quality parameters estimated by the machine learning models with the validation dataset, the optimal model of each water quality parameter for two satellites was selected as follows:

For DO retrieval models, XGBoost had the best performance for Sentinel-2 ($R^2$ = 0.53, MAPE = 22.66%, RMSE = 1.69 mg/L) and ZY1-02D ($R^2$ = 0.53, MAPE = 24.28%, RMSE = 1.67 mg/L) (Figure 8).

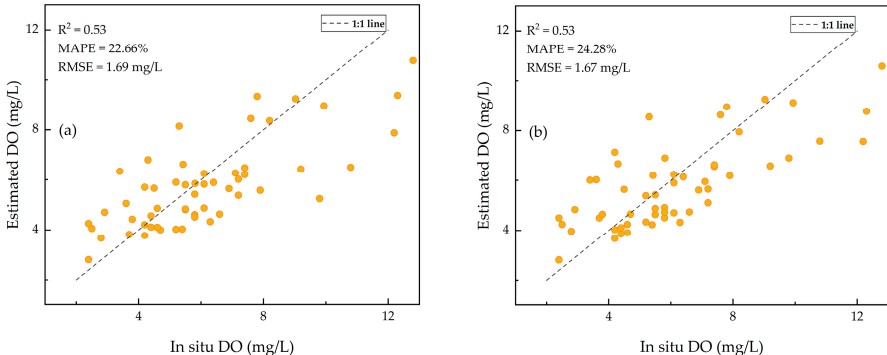

**Figure 8.** Performance evaluation of DO retrievals using the XGBoost for Sentinel-2 (**a**) and ZY1-02D (**b**).

For $COD_{Mn}$ retrieval models, SVR had the best performance for Sentinel-2 ($R^2$ = 0.71, MAPE = 17.96%, RMSE = 0.91 mg/L) and ZY1-02D ($R^2$ = 0.68, MAPE = 18.44%, RMSE = 0.96 mg/L) (Figure 9).

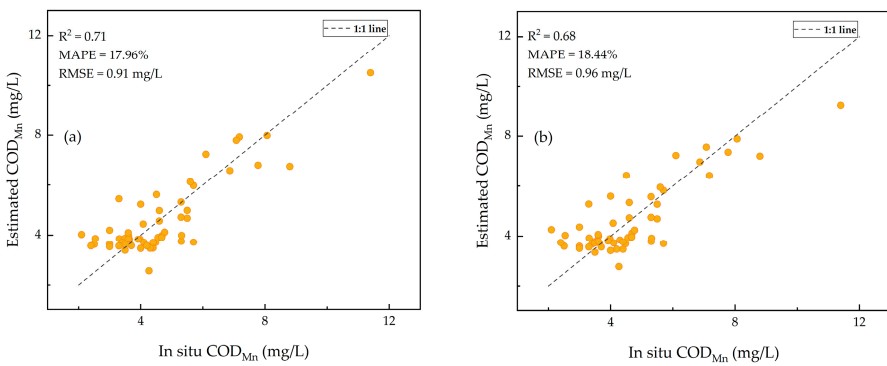

**Figure 9.** Performance evaluation of $COD_{Mn}$ retrievals using the SVR for Sentinel-2 (**a**) and ZY1-02D (**b**).

For TP retrieval models, SVR had the best performance for Sentinel-2 ($R^2$ = 0.46, MAPE = 37.81%, RMSE = 0.08 mg/L) and XGBoost had the best performance for ZY1-02D ($R^2$ = 0.47, MAPE = 37.04%, RMSE = 0.07 mg/L) (Figure 10).

Table 6 shows the performance comparison of all machine learning models. All models did slightly underestimate high concentrations because the training dataset lacks high concentration data.

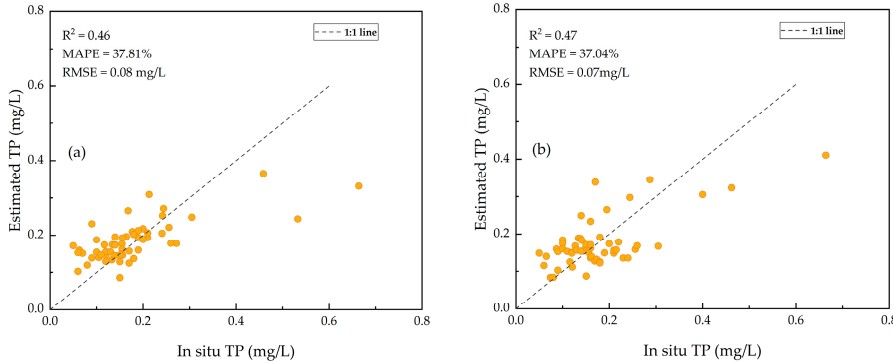

**Figure 10.** Performance evaluation of TP retrievals using the SVR for Sentinel-2 (**a**) and the XGBoost for ZY1-02D (**b**).

**Table 6.** Performance of all machine learning models.

| Parameter | Model | Sentinel-2 | | | ZY1-02D | | |
|---|---|---|---|---|---|---|---|
| | | $R^2$ | MAPE (%) | RMSE (mg/L) | $R^2$ | MAPE (%) | RMSE (mg/L) |
| DO | SVR | 0.43 | 22.12 | 1.85 | 0.39 | 22.18 | 1.92 |
| | PLSR | 0.34 | 28.24 | 1.98 | 0.35 | 27.19 | 1.97 |
| | KNN | 0.33 | 26.42 | 2.00 | 0.31 | 26.68 | 2.03 |
| | XGBoost | 0.53 | 22.66 | 1.69 | 0.53 | 24.28 | 1.67 |
| $COD_{Mn}$ | SVR | 0.71 | 17.96 | 0.91 | 0.68 | 18.44 | 0.96 |
| | PLSR | 0.65 | 18.10 | 1.01 | 0.65 | 17.21 | 1.01 |
| | KNN | 0.65 | 17.11 | 1.00 | 0.66 | 17.03 | 0.99 |
| | XGBoost | 0.58 | 19.53 | 1.10 | 0.65 | 17.99 | 1.0 |
| TP | SVR | 0.46 | 37.81 | 0.08 | 0.36 | 46.97 | 0.086 |
| | PLSR | 0.42 | 37.11 | 0.082 | 0.34 | 37.54 | 0.088 |
| | KNN | 0.43 | 37.18 | 0.082 | 0.46 | 37.67 | 0.079 |
| | XGBoost | 0.39 | 40.88 | 0.079 | 0.48 | 37.04 | 0.073 |

### 3.2.3. Further Validation on Taihu Lake

In situ $R_{rs}$ data from turbid and eutrophic shallow Taihu Lake were used to estimate DO, $COD_{Mn}$, and TP, and were compared with measured data to determine the potential of the machine learning models for producing spatial and temporal products. The $R_{rs}$-derived $COD_{Mn}$ values were similar to the in situ data. However, the $R_{rs}$-derived DO values were less than the in situ data and the $R_{rs}$-derived TP values were more than the in situ data (Figure 11).

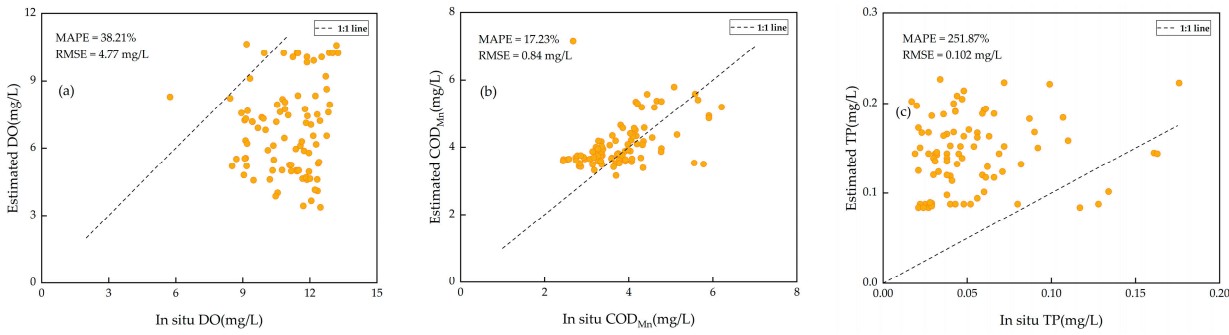

**Figure 11.** Comparison between AHSI $R_{rs}$-derived and in situ DO (**a**), $COD_{Mn}$ (**b**), and TP (**c**).

### 3.3. Water Quality Mapping

The water quality maps derived from ZY1-02D AHSI and Sentinel-2 MSI are shown in Figures 12–14. The waters in the images are Dianshan Lake and the upper reaches of

the Huangpu River which provide more than 60% of the domestic water for residents [48]. Huangpu River is densely populated with boats, making it difficult to retrieve water quality.

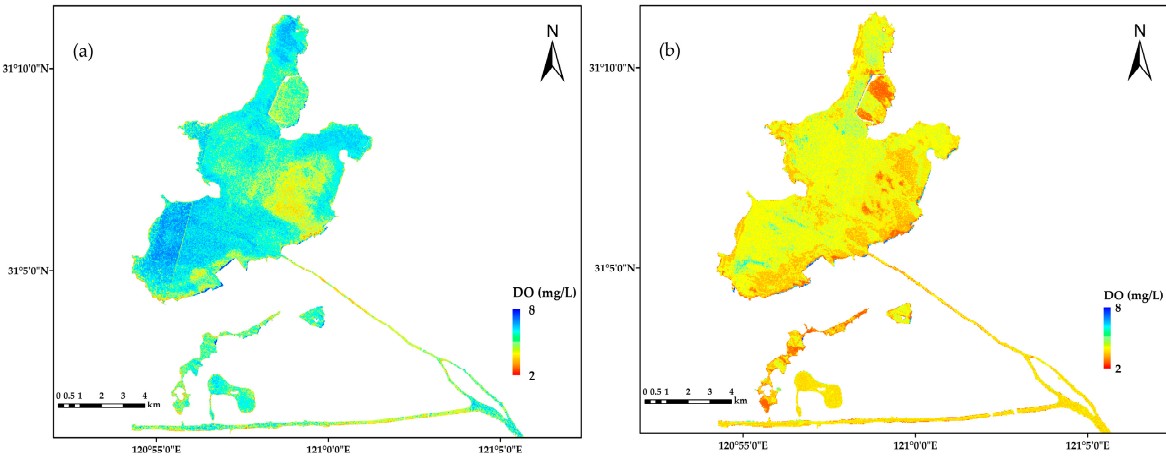

**Figure 12.** DO maps derived from (**a**) Sentinel-2 and (**b**) ZY1-02D images.

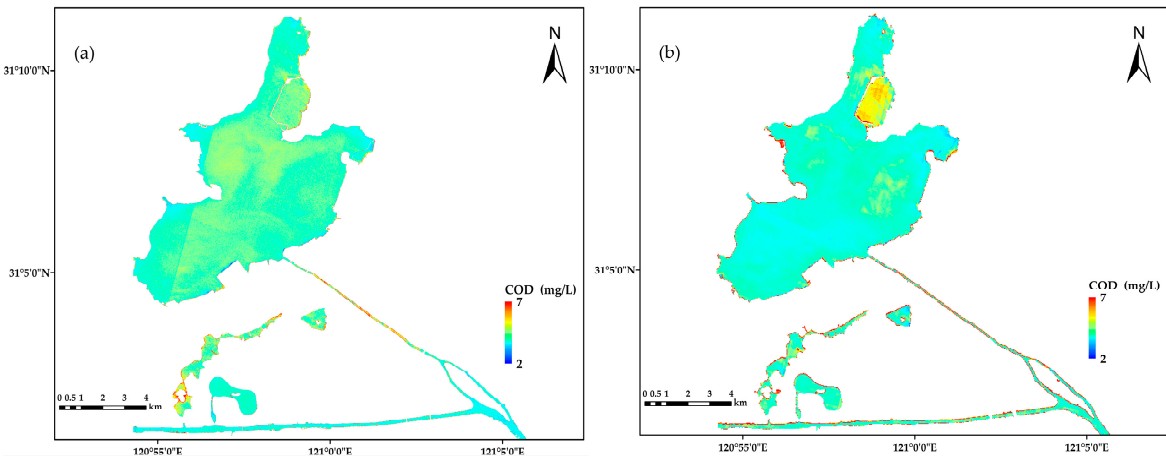

**Figure 13.** COD$_{Mn}$ maps derived from (**a**) Sentinel-2 and (**b**) ZY1-02D images.

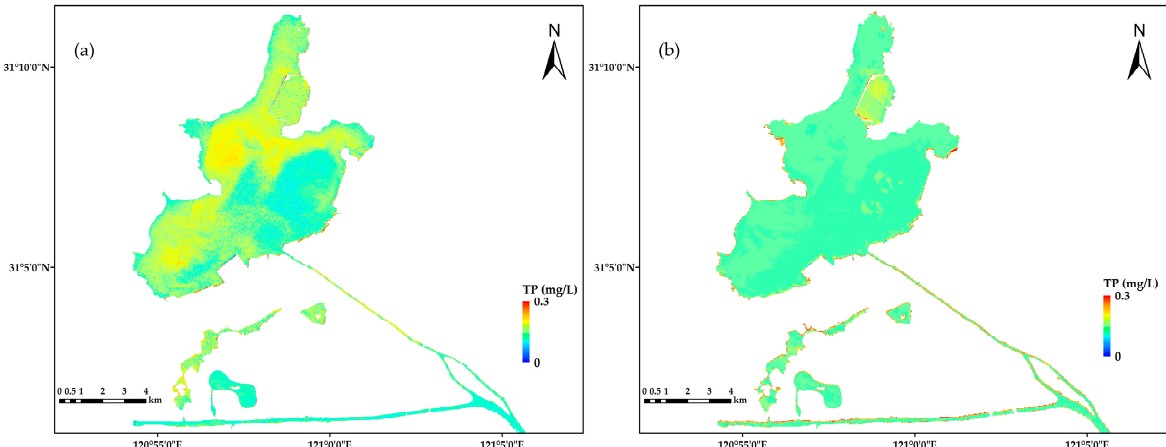

**Figure 14.** TP maps derived from (**a**) Sentinel-2 and (**b**) ZY1-02D images.

The range of the color bar was retained consistently for a better comparison. DO derived from Sentinel-2 and ZY1-02D images had similar spatial distribution characteristics; the DO concentration in the eastern part of Dianshan Lake and the outflow rivers was low

(Figure 12). $COD_{Mn}$ derived from Sentinel-2 and ZY1-02D images are relatively average in the study area, which is consistent with the distribution of the in situ measured value. ZY1-02D overestimated the $COD_{Mn}$ concentration in the aquaculture area in the northeastern part of Dianshan Lake (Figure 13). The agreement is relatively strong for TP retrievals and DO retrievals of Sentinel-2, while the TP derived from the ZY1-02D image showed some limitations and higher errors associated (Figure 14).

### 4. Discussion

#### 4.1. Comparison of $R_{rs}$ Products between ZY1-02D and Sentinel-2

We compared the average $R_{rs}$ of Sentinel-2 and ZY1-02D images at Dataset 2 sites. As shown in Figure 15, the ZY1-02D spectra are in good agreement with Sentinel-2 spectra in terms of both shape and magnitude. However, ZY1-02D $R_{rs}$ are brighter than those of Sentinel-2 in these sites.

The accuracy of the image-derived $R_{rs}$ affected the accuracy of the estimated water quality parameter concentration. Therefore, the agreement evaluation of ZY1-02D and Sentinel-2 image-derived $R_{rs}$ was conducted using $r$ and RMSE (Figure 16). Here, 704 nm had the highest $r$ and lowest RMSE, which was the band with the highest agreement between ZY1-02D $R_{rs}$ and Sentinel-2 $R_{rs}$. Similarly, 560 and 665 nm also had a high agreement. The energies of these bands were higher than that of other wavelengths, and they were relatively less affected by noise. Additionally, 443 and 492 nm were affected by atmosphere scattering. The band after 740 nm was lower in energy; therefore, their agreement was relatively low.

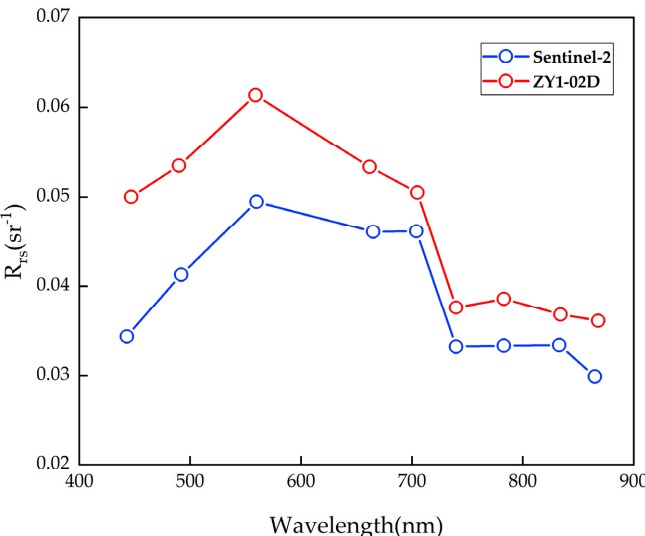

**Figure 15.** Average remote sensing reflectance of Sentinel-2 and ZY1-02D images at Dataset 2 sites.

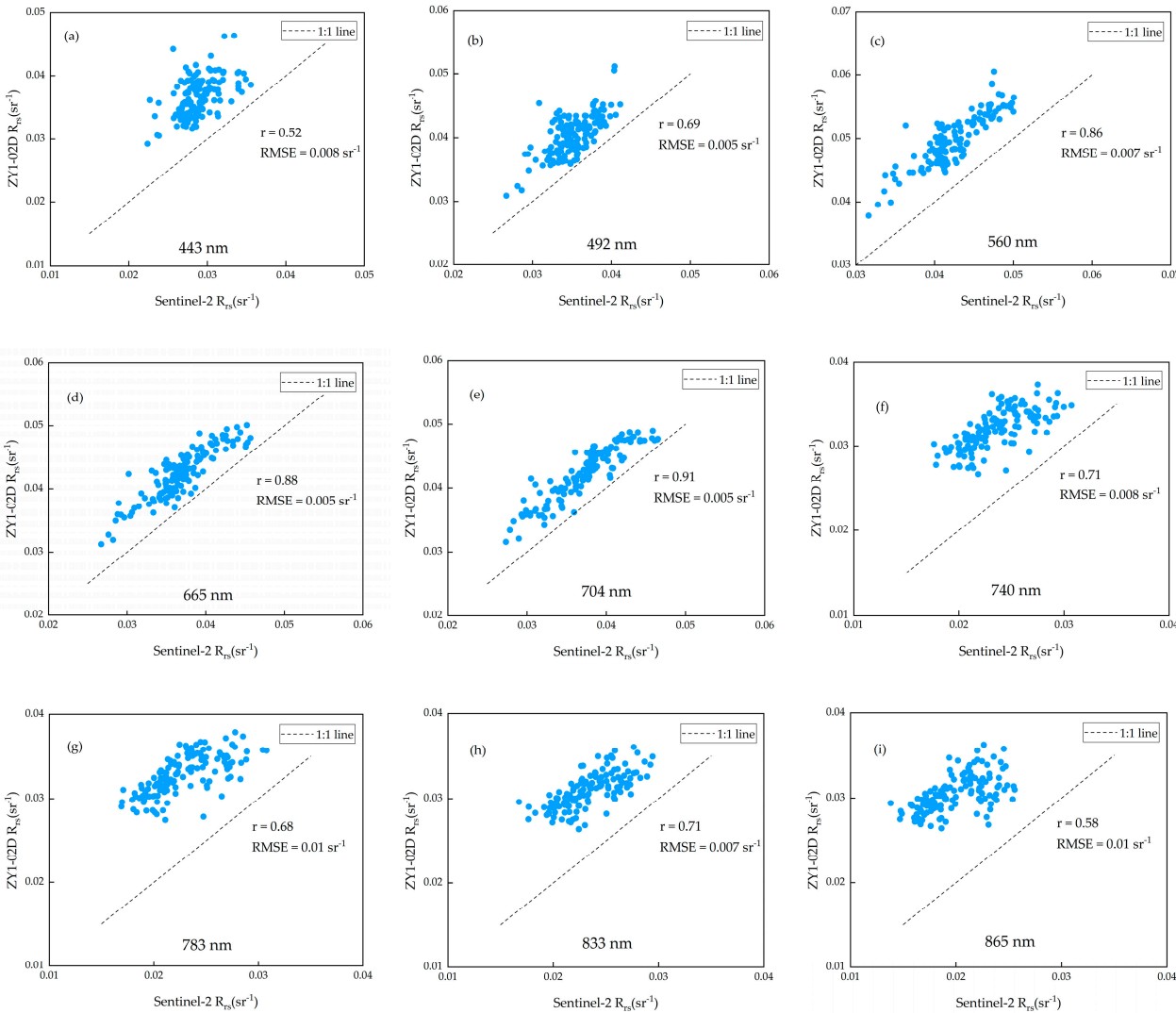

**Figure 16.** Comparison of $R_{rs}$ data between ZY1-02D and Sentinel-2 (**a–i**) represent the band-center wavelength of Sentinel-2, respectively.

### 4.2. Comparison of Water Quality Products between ZY1-02D and Sentinel-2

The biggest challenge of applying models to satellite images was the performance of AC methods [49–51]. The optimal model of each water quality parameter was used to retrieve water quality parameter concentration for Sentinel-2 and ZY1-02D images of the same day. Given the lack of in situ $R_{rs}$ for the direct test of AC, Dataset 2 was used to further validate the accuracy and stability of the models on the satellite images. Spatial windows (3 × 3 pixels) were applied to extract the average concentration at the location of Dataset 2 stations. Figure 17 shows the comparison of concentration between Dataset 2 and image retrievals. The satellite retrievals and the in situ $COD_{Mn}$ had the highest agreement, but they still had significant deviations at high and low concentrations. The DO and TP retrievals of ZY1-02D had large differences with in situ measurements. This can be attributed to the three-day difference between satellite overpasses and in situ measurement as the distribution of the parameters may have some variations.

The comparison of water quality products between ZY1-02D and Sentinel-2 is shown in Figure 18. The agreement of the retrieved concentration is examined by comparing the values extracted from the water quality products. There is a high agreement between the $COD_{Mn}$ retrievals from ZY1-02D and Sentinel-2. Nevertheless, the DO retrievals from ZY1-02D are underestimated compared with those of Sentinel-2. The TP retrievals from ZY1-02D are not successful in this image. TP is the parameter most affected by the atmosphere.

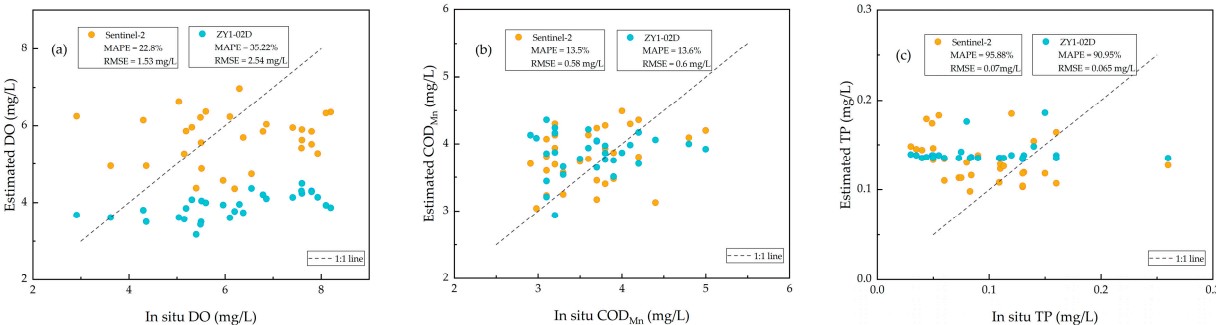

**Figure 17.** Comparison between images derived and Dataset 2 DO (**a**), COD$_{Mn}$ (**b**), and TP (**c**).

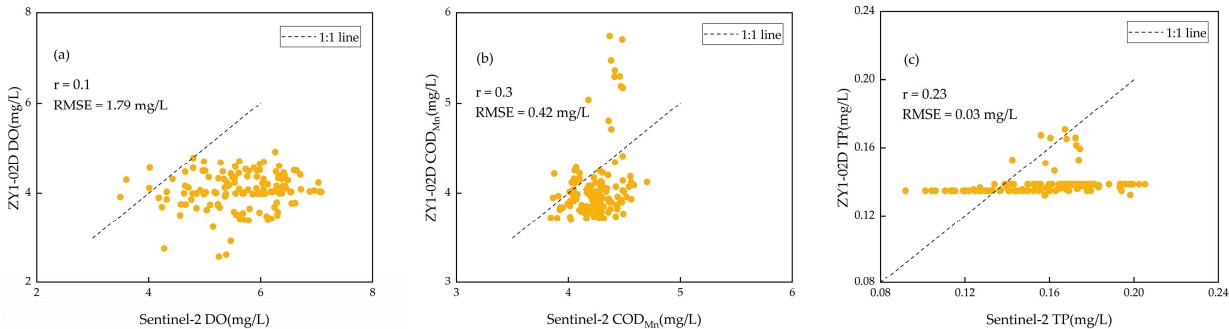

**Figure 18.** Agreement analyses among DO (**a**), COD$_{Mn}$ (**b**), and TP (**c**) derived from ZY1-02D and Sentinel-2 images.

According to the statistical results of accuracy and agreement, even though the models of ZY1-02D and Sentinel-2 trained by in situ data have the same accuracy, the final water quality products are still affected by atmospheric correction, radiometric sensitivity, and signal-to-noise ratio of ZY1-02D and Sentinel-2 sensors.

### 4.3. Strengths and Limitations of the Models

Whether the image-derived modeling methods or in situ observation modeling methods, the applicability of water quality parameter inversion models are limited by the representativeness of the measured data. The revisit time of ZY1-02D is 55 days and the cloud cover is common in Shanghai, resulting in few scenes per year. Our twelve field radiometric measurements surveys included 80 rivers in Shanghai and covered all seasons. For measured rivers, $R_{rs}(550)$ ranged from 0.0103 to 0.079 sr$^{-1}$, which is a relatively broad spectral range. Dataset 1 fully reflects the principal relation between water quality parameters and remote sensing reflectance of the study rivers in Shanghai. Therefore, the models developed by Dataset 1 would be applicable to the water quality parameter retrievals of rivers in the study area though some rivers have not been measured. Compared with the non-optically active water quality parameter (such as TP or TN) retrieval models established by Qiao et al. [10], Gao et al. [18], and Lu et al. [22], their application is limited to specific lakes or rivers and our models have wider applicability. Compared with the DO retrieval model established by Al-Shaibah et al. [16] using a linear model, the DO retrieval model in this study based on XGBoost has higher accuracy.

However, our data did not include water quality in large inland lakes. Here, we used $R_{rs}$ data and water quality data from Taihu Lake in 2009 to examine the further universality of models. The models developed by Dataset 1 had slightly lower performance on data of Taihu Lake than that of the validated dataset described in Section 3.2.2 (Figure 11). This can be attributed to the ten-year difference between the data in Shanghai and Taihu Lake. Moreover, precipitation and human factors are important factors affecting river water quality. Precipitation increases river flow and makes it easier for pollutants to transfer into rivers [52]. The agricultural activities directly transport nutrients into the rivers. These

factors mean the actual retrievals of non-optically active water quality parameters in small rivers may have potential uncertainties.

## 5. Conclusions

In this study, we examined an in situ observation modeling approach based on machine learning for inversion of non-optically active water quality parameters from the hyperspectral ZY1-02D imagery at rivers and lakes in Shanghai. The machine learning models of ZY1-02D have better performance than those of Sentinel-2 because of finer spectral resolution. We conducted analyses on the applicability in different times and spaces of models, and the results showed that the models based on the in situ data in Shanghai can be applied to Taihu Lake. Finally, we validated the accuracy and consistency of $R_{rs}$ and water quality products derived from ZY1-02D compared to those derived from Sentinel-2 images. The comparison of $R_{rs}$ data showed strong agreement at bands of high energies. The $COD_{Mn}$ products showed stronger agreement than DO and TP.

We performed the field-based models' application to a ZY1-02D image which had been atmospherically corrected. However, the FLAASH model used in this study was not for water bodies. Future studies will be dedicated to exploring accurate AC methods for preprocessing ZY1-02D images in the context of quantitative water quality. Overall, the results show the high potential of ZY1-02D hyperspectral imagery in aquatic-oriented applications, though retrieving reliable non-optical water quality parameters is still challenging and further developments are needed.

**Author Contributions:** Conceptualization, C.G. and Y.H.; methodology, Z.Y. and L.L.; formal analysis, Z.Y.; resources, C.G. and T.J.; data curation, Z.Y.; writing—original draft preparation, Z.Y.; writing—review and editing, Z.Y. All authors have read and agreed to the published version of the manuscript.

**Funding:** This research was funded by Shanghai Water Authority Science and Technology Project (Grant No. 2021-10), Science and Technology Commission of Shanghai, Shanghai 2021 "Science and Technology Innovation Action Plan" social development science and technology research project (Grant No. 21DZ1202500), and Jiangsu provincial water resources department, Jiangsu Province Water Conservancy Science and Technology Project (Grant No. 2020068).

**Data Availability Statement:** Not applicable.

**Acknowledgments:** We are also thankful to all anonymous reviewers for their constructive comments provided on the study.

**Conflicts of Interest:** The authors declare no conflict of interest.

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
