# Peer review of "Water Quality Retrieval from ZY1-02D Hyperspectral Imagery in Urban Water Bodies and Comparison with Sentinel-2"

_remotesensing, doi:10.3390/rs14195029_

Round 1
Reviewer 1 Report (Previous Reviewer 1)
Dear author,
You answered my previous comments, but it still leaves the following issues:
Why publish a paper on data that is not publically available, on too little data for good AI training and when you know the AC is not good enough for what you want to apply it to? There exist many publically available ACs for aquatic environments.
I suggest you focus the majority of the paper on your in-situ data.
For that data it is not clear how you processed your observed profile (e.g. the value of \rho(\lambda) for removal of sky light).
I attached an annotated PDF with some additional issues.
All the best
PS
If you feel that I did not understand your paper feel free to contact me and I will be happy to change my review.

Author Response
Response to Reviewer 1 Comments
Point 1: Why publish a paper on data that is not publically available, on too little data for good AI training and when you know the AC is not good enough for what you want to apply it to? There exist many publically available ACs for aquatic environments.
Response 1: Due to weather, economic, and time factors, we only collected 180 in situ data in three years but we think our data has some scientific value. There is no publicly available dataset of measured remote sensing reflectance and non-optically active water quality parameters. If our paper is accepted, then we are willing to share our field data.
As for AC methods, we have tried the acolite and C2RCC methods but had several issues: (1) The dark-spectrum-function method in acolite often fails or is negative in the study area of this paper. (2) Sentinel-2’s water-away reflectance data after C2RCC processing has no results for band7, band8, and band8a. (8) These ACs for aquatic environments don’t currently support ZY1-02D images.
Point 2: For that data it is not clear how you processed your observed profile (e.g. the value of \rho(\lambda) for removal of sky light).
Response 2: We measured the spectrum at a 135° azimuth with respect to the sun and with a nadir viewing angle of 45°. The Fresnel reflectance is assumed to be 0.028 based on the wind speed and sky conditions from field measurements.
Point 3: In the visible, DO absorbs UV light.
Response 3: We overlooked this before, and we added “Although DO can absorb ultraviolet light, this only makes it measurable in the laboratory.”
Point 4: Why do you call your approach semi-empirical? It is fully empirical but implicit. You use an AI to derive an empirical relationship which you don’t know. The other are explicit empirical.
Response 4: In our current knowledge, the empirical methods are defined as training/calibrating a regression model (e.g., polynomial) between image-derived features (e.g., band ratios) and associated concentrations of the constituent of interest known from in situ observations. The semi-empirical methods are built upon training a regression model using in situ observations. Unlike empirical methods for which the training is local (site-specific), the semi-empirical methods are designed as generic models due to the large database of in situ bio-optical measurements used through training the regression model. So our method can be called semi-empirical.
Point 5: Is the dynamics of the parameters you are looking for slow enough that ±3 days is not an issue for matching up?
Response 5: Dataset 2 is mainly distributed in Dianshan Lake and rivers with a width greater than 100m. As far as we know, the water exchange cycle of Dianshan Lake is 7 days. The river flow in the study area is slow and the water quality does not change much in the absence of precipitation.

Reviewer 2 Report (Previous Reviewer 2)
The manuscript can be published after minor revision.
Author Response
Response to Reviewer 2 Comments
Point 1: The manuscript can be published after minor revision.
Response 1: Thank you for acknowledging our work, we have revised the shortcomings of the manuscript. More detailed modifications can be found in the attached pdf.

Reviewer 3 Report (Previous Reviewer 3)
The following comments of my previous review were not correctly addressed:
Then, in the discussion, the authors must compare their results with the obtained results by the other authors for both non-optical and optical parameters. Thus, it would be possible to judge the degree of novelty and accuracy of obtained results.
In the discussion, the authors have to describe the limitation of the methodology. Can this method be applied in other regions? What about the moments in which the water quality changes abruptly (heavy rain periods or organic matter pollution events)?
Author Response
Point 1: In the discussion, the authors must compare their results with the obtained results by the other authors for both non-optical and optical parameters. Thus, it would be possible to judge the degree of novelty and accuracy of obtained results.
Response 1: Thanks for the suggestion, I’m sorry that we overlooked this earlier. We have added section 4.3 “Strengths and Limitations of the Models” to describe these issues in detail. In section 4.3, we first describe the advantages of our method. Since our data are sufficiently representative in time and space, our model is more widely applicable than other authors' models. Machine learning algorithms make our models have higher accuracy.
Point 2: In the discussion, the authors have to describe the limitation of the methodology. Can this method be applied in other regions? What about the moments in which the water quality changes abruptly (heavy rain periods or organic matter pollution events)?
Response 2: In section 4.3, we first described the limitations of our model when applied to Taihu Lake. Then we described the uncertainty that precipitation and human activity bring to model application.

Reviewer 4 Report (Previous Reviewer 4)
Water Quality Retrieval From ZY1-02D Hyperspectral Imagery in Urban Water Bodies and Comparison with Sentinel-2
Dear Authors
The basic science of this paper is conducted in a good way and is of an appropriate standard. The author and his team write this paper according to journal scope and modern trends. I am glad to review this paper because the paper is very interesting according to my research interest area. This manuscript needs more attention. At this stage, I can recommend a major revision due to technical issues. The author should revise the whole manuscript and fix some technical issues with major comments. All figures are not according to journal criteria. I hope, the author will follow our comments and enhance their own study and resubmit again in this journal.
 I agreed about the title of the manuscript but the author should use in low case: Fromïƒ from
 Line 62: In previous studies, Which studies, add references of studies
 There is some typo error. The author should double-check the whole manuscript.
 Figure 1 is not appropriate and according to journal criteria. The author should revise figure 1.
 Why the author used the Y1-02D AHSI, because this satellite data also have a 30m spatial resolution just like Landsat data.
 Most important both types of optical data have different spatial resolutions, how to fix them?
 Line 219-227: there are many references are missing.
 Figure 7 is not according to journal criteria
 There are many figures. The author should add some figures in the supplementary files.
 I hope the authors will revise this manuscript and resubmit it as soon as possible.
I'll follow just my comments.
Best Regards
Author Response
Point 1: I agreed about the title of the manuscript but the author should use in low case: From from
Response 1: We have changed the title to “Water Quality Retrieval from ZY1-02D Hyperspectral Imagery in Urban Water Bodies and Comparison with Sentinel-2”
Point 2: Line 62: In previous studies, Which studies, add references of studies.
Response 2: We added three references to review the researchers using empirical methods to estimate non-optically active parameters.
Point 3: There is some typo error. The author should double-check the whole manuscript.
Response 3: Thank you for your reminder, We double-checked the manuscript and corrected some typo errors
Point 4: Figure 1 is not appropriate and according to journal criteria. The author should revise figure 1.
Response 4: We have added a compass, scale bar, and latitude and longitude lines in Figure 1a.
Point 5: Why the author used the ZY1-02D AHSI, because this satellite data also have a 30m spatial resolution just like Landsat data.
Response 5: The ZY1-02D satellite was successfully launched in September 2019, carrying the new-generation AHSI, which has the same spatial resolution and swath width as GF-5. However, to improve the signal-to-noise ratio of the data, the spectral resolution in the VNIR and SWIR bands of ZY1-02D AHSI was reduced to 10 nm and 20 nm, respectively. As a result, while maintaining wide swath and coverage capability, the signal-to-noise ratio (SNR) of the ZY1-02D AHSI sensor was improved compared with the GF-5 AHSI. The minimum SNR of the sensor under typical operating conditions exceeds 120, allowing for uninterrupted long strip imaging. Therefore, ZY1-02D hyperspectral data have a high potential for application in the quantitative information extraction of inland water bodies.
Point 6: Most important both types of optical data have different spatial resolutions, how to fix them?
Response 6: ZY1-02D has 30m spatial resolution and Sentinel-2 has 10m/20m/60m spatial resolution. First, Sentinel-2 data of different spatial resolutions were all resampled to 10m in preprocessing. Then, when extracting values on Sentinel-2 images, we used a 3*3 spatial window to extract the average values at the location of stations.
Point 7: Line 219-227: there are many references are missing.
Response 7: We also added three references in Line 219-227.
Point 8: Figure 7 is not according to journal criteria.
Response 8: We would like to know more details of the error, but we modified the legend and line colors of Figure 7 anyway.
Point 9: There are many figures. The author should add some figures in the supplementary files.
Response 10: Thank you for your suggestion, but we did not find the link location to add figures in the current interface.

Round 2
Reviewer 1 Report (Previous Reviewer 1)
Dear authors,
Thank you for your answers.
I have issues with two of your answers:
1. you write: Unlike empirical methods for which the training is local (site-specific), the semi-empirical methods are designed as generic models due to the large database of in situ bio-optical measurements used through training the regression model.
This is not the difference between empirical and semi-empirical in my community. Empirical is every approach that rely solely on data (no underlying theory) to derive a relationship between variable. It has nothing to do with the size of the database nor its geographical extent.
2. You write: Dataset 2 is mainly distributed in Dianshan Lake and rivers with a width greater than 100m. As far as we know, the water exchange cycle of Dianshan Lake is 7 days. The river flow in the study area is slow and the water quality does not change much in the absence of precipitation.
Make sure, in the paper, that you highlight this assumption.
Good luck and all the best.
Author Response
Point 1: This is not the difference between empirical and semi-empirical in my community. Empirical is every approach that rely solely on data (no underlying theory) to derive a relationship between variable. It has nothing to do with the size of the database nor its geographical extent.
Response 1: Thanks for your suggestion, we recognize the inadequacy of our knowledge in this area. In the latest version of the manuscript, we have replaced “empirical methods” and “semi-empirical methods” with “image-derived features modeling methods”and “in situ observations modeling methods”. We summarize most of the previous studies by scholars as modeling methods based on image-derived features, because they directly established the relationship between remote sensing image reflectance and water quality parameters. We call the approach used in our paper a modeling approach based on in situ observations, because we establish relationships between measured reflectance and water quality parameters and apply them to remote sensing images.
Point 2: Make sure, in the paper, that you highlight this assumption.
Response 2: In section 2.2.1, we addded “Dataset 2 is mainly distributed in Dianshan Lake and rivers with a width greater than 100m. According to our field investigation, the water change cycle of Dianshan Lake is 7 days and the flow rate of these rivers is slow. Therefore, in the absence of precipita-tion and sudden pollution, the water quality of these sites does not change much in a short period of time.”

Reviewer 4 Report (Previous Reviewer 4)
At this stage. I agree about all changes in the revised manuscript. The author addressed my all suggested comments in a better way.
Best of luck with your mansucript.
Author Response
Thank you for acknowledging our previous revisions, your previous comments made our paper better.

This manuscript is a resubmission of an earlier submission. The following is a list of the peer review reports and author responses from that submission.
Round 1
Reviewer 1 Report
Dear authors,
While I appreciate the need to derive water quality parameters, your approach does not seem to be fruitful for reasons that seems obvious to me. I would expect you first to make the case why you expect your approach to work (otherwise why not use other parameters such as human density around the water body and rain that may be better correlated).
I have several comments that, if addressed, I believe will significantly improve this paper.
1. Too little data is available to run an AI. Typically, much larger data sets are used.
2. R^2 is a poor metric of goodness of fit. I would have liked to see the relative error and absolute error statistics highlighted as well.
3. The Chinese satellite data is not available to the general public. Should not be used unless shared.
4. Why not use an atmospheric correction scheme that is suited for water and why not validate its use by comparing remotely sensed derive Rrs with in-situ.
5. Why do you think you should be able to derive parameters not related to color from color? The only reason is that there is a link between parameters related to color (absorption by non-algal particles, dissolved materials and algae and scattering by particles) to those that are not and you are trying to derive. This link needs to be SPELLED OUT. Otherwise one wonders about your ability to generalize your result to the next year, let alone to another location.
6. SST is also linked to many in-water processes, and particularly DO solubility. Shouldn't it be used as well (available, at water surface, from Landsat 8/9)?
7. Any time you add a wavelength you can expect to improve the fit (just like increasing the order of a polynomial to make it pass better between points). To make the case that you are not just fitting noise, you need to show that your data points are truly independent.
Dear authors, I am often wrong. If you feel my comments are 'off the mark' feel free to contact me and if convinced I will be happy to change my review.
Emmanuel Boss, UMaine
Reviewer 2 Report
The main steps in this paper including the following steps:
Step 1) collecting in situ data, including spectral reflectance, R_rs, and water quality data, DO, CODMn, TP. The final number of datasets is 183.
Step 2) calculating the simulated satellite data by convoluting the above R_rs with the spectral response function (SRF) of multispectral (Sentinel-2) and hyperspectral (ZY1-02D) sensors.
Step 3) developing models with semi-empirical and machine learning models using the simulated satellite data and the water quality data.
Step 4) apply the above models on the real satellite data.
So the main comparison between multispectral and hyperspectral sensor in monitoring water quality is based on the accuracy comparing on those models developed in Step 3). The spectral used to develop the models are calculated by the in situ spectral reflectance and SRF of multi and hyperspectral sensors. Since the in situ spectral data are the same, so the differences are actually caused by the SRFs of the sensors.
Yet, the actual satellite data has a much large difference then the simulated data. So, are the above comparison results derived from the simulated data also correct in real remote sensing application? Can the models derived from simulated data be applicable on real satellite data? I think those issues will be more interested to readers.
Reviewer 3 Report
The authors have presented the development of a correlation model based on remote sensing to estimate the presence of three non-optical parameters of water quality. The authors report their results and validation of the model. In general terms, the paper is well written and well structured. Nonetheless, some aspects must be enhanced in order to reach the necessary quality to be published in Remote Sensing. Following, I have included some of the aspects to be improved.
In Figure 4, it is necessary to modify the axis to add the wavelengths instead of the name of the band. Apply this comment for other similar cases, such as Figure 5.
Table 4 must be enhanced. The correlation value is a good indicator of the correlation, but other parameters such as p-value and MAE/MAPE must be included for each correlation.
The MAE/MAPE and other parameters should be given and analyzed to evaluate the verification process.
In order to evaluate the novelty of the proposed methodology, the authors must compare their results with the existing literate. Therefore, two aspects should be addressed. A related work section has to be added. In this section, the authors will provide a comprehensive review of current literature in the field of remote sensing for non-optical parameters. Then, in the discussion, the authors must compare their results with the obtained results by the other authors for both non-optical and optical parameters. Thus, it would be possible to judge the degree of novelty and accuracy of obtained results.
In the discussion, the authors have to describe the limitation of the methodology. Can this method be applied in other regions? What about the moments in which the water quality changes abruptly (heavy rain periods or organic matter pollution events)?
At the end of the conclusions, the authors must shortly describe the future work linked to this research.
Reviewer 4 Report
Dear Authors
Comparison of Multispectral and Hyperspectral Sensor in Monitoring Shanghai Water Quality using Sentinel-2 and ZY1- 02D Imagery Dear Authors
I am glad to review this manuscript. The author compared the multispectral and hyperspectral remote sensing data for the monitoring of water quality in shanghai using S-2 and Zy1-02D imagery data. This manuscript is according to journal scope and they presented in very good form. I found some minors problem in this study. I recommend minor revisions.
Find to attached some comments are given below.
Minor Comments
1. Line 17-20: it's messy. Revise this sentence
2. Add some results in the abstract section
3. Line 60: In the previous studies; add some references here.
4. Figure 1 is not appropriate. Where shanghai exists. The author explained the first shanghai exist in China. The author prepared two maps. Figure 1a explains china and Figure 1b explains shanghai.
5. Provide a link, where you get in situ data.
6. Try to combine figures 10, 11, 12 to 10a, 10b, and 10c (just like figure 9)
In the end, I would like to say about your study. I believe you did a great job but we still need some improvement in your paper. I hope you will modify it very soon and resubmit it again in this journal.
Best Regards